# Does the Use of the “Proseek^®^ Multiplex Inflammation I Panel” Demonstrate a Difference in Local and Systemic Immune Responses in Endometriosis Patients with or without Deep-Infiltrating Lesions?

**DOI:** 10.3390/ijms24055022

**Published:** 2023-03-06

**Authors:** Alexandra Perricos, Heinrich Husslein, Lorenz Kuessel, Manuela Gstoettner, Andreas Weinhaeusel, Thomas Eiwegger, Gabriel Beikircher, René Wenzl

**Affiliations:** 1Department of Obstetrics and Gynecology, Medical University of Vienna, 1090 Vienna, Austria; 2Molecular Diagnostics, Center for Health & Bioresources, AIT Austrian Institute of Technology Vienna, 1210 Vienna, Austria; 3Translational Medicine Program, Research Institute, The Hospital for Sick Children, Toronto, ON M5G 1X8, Canada; 4Departments of Pediatric, Temerty Faculty of Medicine, University of Toronto, Toronto, ON M5S 1A8, Canada; 5Departments of Immunology, Temerty Faculty of Medicine, University of Toronto, Toronto, ON M5S 1A8, Canada; 6Medical Faculty, Karl Landsteiner University of Health Sciences, 3500 Krems an der Donau, Austria; 7Department of Pediatric and Adolescent Medicine, University Hospital St. Pölten, 3100 St. Pölten, Austria

**Keywords:** endometriosis, inflammation, multiplex analysis

## Abstract

Endometriotic lesions are able to infiltrate surrounding tissue. This is made possible partly by an altered local and systemic immune response that helps achieve neoangiogenesis, cell proliferation and immune escape. Deep-infiltrating endometriosis (DIE) differs from other subtypes through the invasion of its lesions over 5 mm into affected tissue. Despite the invasive nature of these lesions and the wider range of symptoms they can trigger, DIE is described as a stable disease. This elicits the need for a better understanding of the underlying pathogenesis. We used the “Proseek^®^ Multiplex Inflammation I Panel” in order to simultaneously detect 92 inflammatory proteins in plasma and peritoneal fluid (PF) of controls and patients with endometriosis, as well as in particular patients with DIE, in order to gain a better insight into the systemically and locally involved immune response. Extracellular newly identified receptor for advanced gycation end-products binding protein (EN-RAGE), C-C motif Chemokine ligand 23 (CCL23), Eukaryotic translation initiation factor 4—binding protein 1 (4E-BP1) and human glial cell-line derived neurotrophic factor (hGDNF) were significantly increased in plasma of endometriosis patients compared to controls, whereas Hepatocyte Growth factor (HGF) and TNF-related apoptosis inducing ligand (TRAIL) were decreased. In PF of endometriosis patients, we found Interleukin 18 (IL-18) to be decreased, yet Interleukin 8 (IL-8) and Interleukin 6 (IL-6) to be increased. TNF-related activation-induced cytokine (TRANCE) and C-C motif Chemokine ligand 11 (CCL11) were significantly decreased in plasma, whereas C-C motif Chemokine ligand 23 (CCL23), Stem Cell Factor (SCF) and C-X-C motif chemokine 5 (CXCL5) were significantly increased in PF of patients with DIE compared to endometriosis patients without DIE. Although DIE lesions are characterized by increased angiogenetic and pro-inflammatory properties, our current study seems to support the theory that the systemic immune system does not play a major role in the pathogenesis of these lesions.

## 1. Introduction

Endometriosis is a benign gynecological disease, defined as the implantation of endometrial-like tissue outside of the uterine cavity [1], occurring with a prevalence of 6–10% in women of reproductive age [2]. Its clinical presentation varies from asymptomatic women to patients suffering from dysmenorrhea, dyspareunia, dyschezia, dysuria and/or infertility [1]. Due to this range of possible symptoms, diagnosis is often delayed, in many cases taking a great toll on the quality of life of affected women [3]. The unacceptably long time to diagnosis, which has reported to be over eight years, on average, for women suffering from pelvic pain [4], calls for a better understanding of the disease itself and new, non-invasive biomarkers.

Depending on the locations and characteristics of the lesions, three endometriosis subtypes have been defined: (superficial) peritoneal lesions, ovarian endometriosis cysts (known as endometriomas) and deep-infiltrating endometriosis (DIE), defined as an infiltration of ≥5 mm into the surrounding tissue, such as, more commonly, the uterosacral ligaments, the bladder or the rectum [5]. Due to its wide range of symptoms, this endometriosis subtype is known to potentially have a severe impact on patient’s quality of life [6]. This underlines the importance of an adequate choice of treatment. Treatment options include a conservative as well as a surgical approach. The choice should be based on the patient’s medical history, symptoms, extent of the disease and personal preference [7]. Despite the invasive nature of these lesions and the wider range of symptoms they can trigger, DIE is described as a stable disease [8].

The exact pathophysiology of this disease still remains unknown. Numerous studies over the years have described inflammation as a key feature in the pathogenesis of endometriosis. Endometriotic lesions are characterized by their ability to implant and proliferate [9,10], thus infiltrating the surrounding tissue [11], triggering neoangiogenesis and an invasion of nerve fibers [12]. Important for their survival is not only their resistance to apoptosis, but also their escape from immune surveillance [13,14]. The first line of defense against any aberrant cells, be it tumor cells or virus-infected cells, are, amongst others, natural killer cells (NK-cells), macrophages, granulocytes and mast cells. These innate cells secrete chemokines (notably from the CC- and CXCL family) and cytokines which recruit additional leukocytes and form the basis for chronic inflammation [15]. In addition, also non hematopoietic cells can contribute to inflammation. This partially explains why high concentrations of these inflammatory proteins have been observed in context with low concentration of NK-cells in the peritoneal milieu of women with endometriosis [14]. Chronicity and lack of clearance of inflammation may be maintained by an impairment of the phagocyting capacities of attracted macrophages [15]. 

A simultaneous detection of different proteins involved in inflammatory processes has been made possible by multiplex technologies. Several studies have evaluated the usefulness of multiplex technologies for the detection of proteins associated with endometriosis [16,17]. While the analysis of inflammatory proteins in plasma may offer information about the systemic conditions brought on by the disease, the analysis of peritoneal fluid (PF) has established itself as a useful tool to learn more about the local environment of pelvic endometriotic lesions. 

In this study, we used the “Proseek^®^ Multiplex Inflammation I Panel” in order to expand our understanding of the local intraperitoneal as well as systemic inflammatory mechanisms that underlie the pathogenesis of endometriosis. First, we sought to compare patients suffering from endometriosis to controls to define differences in the immune response brought on by the disease itself. Since characteristics of the endometriosis subtypes differ from each other, we then attempted to define distinct local and systemic inflammatory properties of patients with deep-infiltrating endometriotic lesions.

## 2. Results

Eighty-four patients were included in our study. In three of these study participants, however (one patient with endometriosis without DIE, one patient with DIE and one control), ≥50% of markers were found to be under the limit of detection (LOD), either in plasma or in PF. These three patients were therefore excluded from further analysis. Thus, 81 patients were included in the final calculations, 51 patients with endometriosis (18 of which suffered from DIE) and 30 controls. Patient characteristics are summarized in Table 1.

For further data evaluation, we disregarded 23 (underlined in the Appendix A) out of the 92 analyzed proteins in PF, because they were expressed in less than 50% of patients. The same was carried out for 16 proteins (set in bold in the Appendix A) measured in plasma.

### 2.1. Patients with Endometriosis vs. Controls

The first step of this study focused on comparing all patients with endometriosis to controls, regarding local and systemic inflammatory response. Table 2 presents the markers that differed significantly between the two groups, in both plasma and PF. Heat maps of these proteins are shown in Figure 1 (plasma) and Figure 2 (PF).

In plasma, four proteins were significantly increased in endometriosis patients, notably the Extracellular Newly identified Receptor for Advanced Glycation End-products binding protein (EN-RAGE—1.63-fold), Chemokine ligand 23 (CCL23—1.22-fold), Eukaryotic translation initiation factor 4E-binding protein 1 (4E-BP1—1.43-fold) and human Glial-derived Neurotrophic factor (hGDNF—1.12-fold), whereas Hepatocyte Growth Factor (HGF) and TNF-Related Apoptosis Inducing Ligand (TRAIL) were significantly decreased in this patient group.

When analyzing PF, we merely found three protein concentrations that significantly differed between the groups: Interleukin-6 (IL-6) and Interleukin-8 (IL-8) were increased (2.13- and 2.68-fold respectively), while Interleukin-18 (IL-18) was decreased 0.67-fold in PF of women suffering from endometriosis, compared to controls. 

### 2.2. Patients with DIE vs. Patients with Endometriosis without DIE

In order to determine the influence of a deep-infiltrating lesion on the local and systemic inflammatory milieu, we compared the inflammation markers in plasma and PF of patients with DIE to women with endometriosis but without DIE. The proteins that differed significantly between these groups are summarized in Table 3. Heat maps of these proteins are shown in Figure 3 (plasma) and Figure 4 (PF). 

In plasma, two proteins were significantly decreased in patients with DIE. Tumor necrosis factor-related activation-induced cytokine (TRANCE) and chemokine (C-C Motif) ligand 11 (CCL11) were decreased 0.70- and 0.84-fold, respectively, in patients with DIE. In PF four proteins were increased in patients with deep-infiltrating lesions: CCL23 (1.7-fold), CCL11 (1.46-fold), Stem Cell Factor (SCF—1.23-fold) and C-X-C motif chemokine 5 (CXCL5—2.54-fold).

## 3. Discussion

The key role of inflammatory processes in the development of endometriosis and its lesions has been evaluated in multiple studies. While many theories have been discussed regarding the pathogenesis of endometriosis, many questions still remain to be answered. In our study, we used a multiplex technology which enabled a simultaneous detection of inflammatory proteins in PF and plasma of patients suffering from endometriosis and controls, in hopes of gaining new insights into underlying pathophysiological mechanisms of endometriosis, and especially of DIE. 

We therefore performed two sub-analyses in our study, comparing plasma and PF of patients with endometriosis to samples of controls, and in a second step, comparing patients suffering from DIE to endometriosis patients without deep-infiltrating lesions. 

### 3.1. Patients with Endometriosis vs. Controls

In Table 4 we summarize the main characteristics of proteins that were significantly increased or decreased in patients suffering from endometriosis compared to controls.

Two markers (EN-RAGE and CCL23) known for their chemoattractant characteristics were found to be increased in plasma of endometriosis patients compared to controls.

EN-RAGE genes were found to be significantly higher expressed in endometrial stromal cells (ESC) in endometriosis patients compared to controls by Sharma et al. [32]. EN-RAGE expression seems to be upregulated by its own receptor, RAGE, the expression of which might in turn be activated by estradiol. This led to the conclusion that EN-RAGE might play an important role in the altered inflammatory response in endometriosis patients. Interestingly, an increased expression of EN-RAGE in ESC as described by Sharma would consequently result in an increased expression in PF. Our data, however, merely showed a significant systemic and not local increase in EN-RAGE, as it was increased in plasma and not in PF. CCL23 has never, to our knowledge, been associated with endometriosis in previous studies. Our analysis, however, showed a significant systemic increase in endometriosis patients.

The association between HGF and endometriosis has been described in numerous studies [7], promoting cell proliferation and angiogenesis [22], two central characteristics of endometriotic lesions. Most studies, however, focused on the local HGF production, measured in PF, rather than on its systemic expression. Recently, Zhang et al. showed a significantly increased expression of HGF in serum of patients suffering from endometriosis compared to controls, and in particular an increased expression in endometriosis stages III and IV, suggesting that this protein might be used as a future biomarker for diagnosing more advanced stages of this disease [33]. In contrast, we found this protein to be significantly decreased in plasma of patients suffering from various stages of endometriosis. This puts into question the validity of HGF as a future biomarker for this disease. 

Gene expression of 4EBP1was found to be increased in eutopic endometrium of endometriosis patients yet decreased in ovarian endometriotic lesions in these patients [34]. These changes could not, however, be sufficiently explained. 

The main function of TRAIL lies in its ability to induce apoptosis. Kim et al. showed results similar to those found in our analyses, with a reduced TRAIL expression in serum of women suffering from endometriosis [35], a trait also seen in some types of cancer cells [26]. This downregulated TRAIL expression might play a significant role in the resistance of endometriotic lesions to apoptosis.

Very little data have been published to date regarding the involvement of hGDNF in endometriotic lesions. Chang et al. and Streiter et al. demonstrated the importance of this neurotrophic factor in normal ovarian function and follicular development and suggested that an altered expression might lead to the development of ovarian diseases such as polycystic ovary syndrome (PCOS) or endometriosis [36,37]. To our knowledge, the only study that evaluated a direct link of this protein to endometriosis showed an increased expression in the glandular epithelium of peritoneal ectopic lesions compared to stroma and surrounding peritoneum [38]. 

In PF of endometriosis patients compared to controls, three pro-inflammatory cytokines (IL-6, IL-8 and IL-18) were significantly differently expressed. The data published regarding IL-18 expression in endometriosis patients show discrepant results. Our results are in line with those described by Zhang et al.: Here too, IL-18 expression was significantly reduced in PF of patients with endometriosis, while this did not correlate with expression in plasma. This finding was used to explain the known reduced NK-cell activity and impaired T-helper-cell immune response seen in patients with endometriosis [39]. In contrast, Oku et al. reported no differences in circulating IL-18 concentrations between these two groups, however, a significantly increased expression in PF of affected patients [40], while Glitz et al. described no differences in serum or PF regarding IL-18 expression but showed a strong positive correlation between serum and PF IL-18 levels [41]. 

On the other hand, IL-6 and IL-8 were overexpressed in PF of women suffering from endometriosis, which underlines the local pro-inflammatory environment seen in our patient collective. Our data confirm the results found by Wang et al., who described a proliferation-stimulating effect on peritoneal endometriotic lesions [42]. Li et al. further postulated that the increased IL-6 expression induced by macrophages increases secretion of haptoglobin, which in turn helps endometriotic cells escape immune surveillance by binding to macrophages and thus decreasing phagocytosis [43]. As shown in the review by Sikora et al., the increased expression of Il-8 is thought to be linked to an increased expression of other pro-inflammatory factors such as IL-1 and TNF-alpha in patients with endometriosis [31]. Furthermore, it was shown that IL-8 production in human endometrial endothelial cells (HEEC) is stimulated by estrogen and progesterone in women with endometriosis, while these steroid hormones do not have the same effect on HEEC in women without endometriosis [44]. 

### 3.2. Patients with DIE vs. Endometriosis Patients without DIE

Table 5 summarizes the main characteristics of proteins that were significantly increased or decreased in patients with DIE compared to endometriosis patients without DIE.

DIE differs from other endometriosis subtypes in that these lesions penetrate exceeding 5 mm into surrounding tissue. Because of its particular characteristics, the understanding of the pathophysiology of this subtype has become the center of recent studies. Although DIE should not be considered a progressive disease, the impact on patient’s symptoms and the challenges for surgeons dealing with these lesions are high [8]. 

Zhou et al. recently described different cytokine signatures differentiating between the three previously mentioned subtypes of the disease (peritoneal, ovarian and deep-infiltrating endometriosis). The authors used a multiplex assay of 48 cytokines on PF of endometriosis patients and found a six-cytokine signature of IL-8, IL-12p70, IL-16, IL-18, MCP-1 and MIP-1α, all upregulated in patients with DIE compared to peritoneal endometriosis, as well as a seven-cytokine signature comprising IL-1α, IL-1RA, IL-8, IL-12p40, IL-12p70, IL-16 and TNF-α when comparing patients with ovarian endometriomas and patients with DIE [51]. Although several of these cytokines (IL-8, IL-18, MCP-1 and IL-1α) were included in our analysis, we could not find any corresponding results in our samples.

Compared to endometriosis patients without DIE, two proteins were found to be significantly decreased in plasma of patients with DIE. 

To our knowledge, TRANCE has never been connected to the pathophysiology of endometriosis. Our data show lower values in plasma of patients suffering from DIE, which might contribute to immune escape seen in endometriotic lesions.

The important role of CCL11 in endometriotic lesions has been attributed to its recently described angiogenetic potency [52]. In our collective, we found the chemoattractant CCL11 to be diminished in plasma of patients suffering from DIE yet increased in PF of these patients. While in our samples CCL11 concentrations merely differed in patients with DIE compared to endometriosis patients without DIE, previously published data showed lower values in serum [53] and increased values in PF of endometriosis patients compared to controls [26], which correlated with the endometriosis rASRM stage.

Suzumori et al. have described significantly elevated levels of CXCL5 in PF of endometriosis patients compared to controls, in particular, however, in patients suffering from more severe stages (stage III and IV) [54]. While our data merely showed differences between DIE and patients without DIE, this may be attributed to the fact that we had significantly more patients with moderate to severe endometriosis in the DIE group. Furthermore, Wunder et al. reported an increased concentration of the chemokine in follicular fluid (FF) of patients with endometriosis. As FF contributes to PF during ovulation, this might add to the inflammatory milieu as well as implantation and neovascularization of endometriotic lesions [55]. An overexpression of CXCL5 has also been previously described in deep-infiltrating lesions themselves, notably in rectovaginal lesions [56].

Another cytokine involved in neoangiogenesis is the previously mentioned CCL23. This cytokine was found to be significantly overexpressed in our patient collective suffering from DIE compared to endometriosis patients without DIE, which might suggest an increased vascularization of these lesions.

Osuga et al. analyzed the concentration of SCF in plasma and PF of patients with endometriosis and controls. The authors described an increased expression of SCF only in PF of affected patients. When comparing patients according to their endometriosis rASRM stage, this significant difference was only maintained in patients with stages I and II, compared to controls [49]. Here, it is important to consider, however, that the rASRM score solely takes into account intraperitoneal lesions and does not offer any insight into the extent of possible deep-infiltrating lesions. In our analysis, we found significantly higher SCF concentrations in PF samples of patients with DIE compared to endometriosis patients without DIE. This reconfirms the role of SCF in local inflammation.

In vivo data have shown that there seems to be a causal effect of a systemic immune response on the development of endometriotic lesions [57]. However, further studies are needed to clarify whether the role of the immune systems is truly causal or merely a consequence of this disease.

In conclusion, while the important role of inflammatory processes in the development and persistence of endometriotic lesions has been repeatedly demonstrated, our multiplex analysis of 92 inflammatory proteins failed to show drastic differences regarding local and systemic inflammatory response in patients suffering from endometriosis compared to controls. Surprising to us were especially the very few differences found in patients with deep-infiltrating lesions. As such, we believe it may be difficult to identify biomarkers that allow the non-invasive diagnosis of DIE. Although these lesions invade deeper into surrounding tissue and create more anatomic distortions, adhesions and tissue fibrosis, our data showed only few differences regarding inflammatory marker levels measured in the “Proseek^®^ Multiplex Inflammation I Panel” in patients suffering from DIE. Although DIE is a subtype characterized by increased angiogenetic and pro-inflammatory properties [58], our current study seems to support the theory that the systemic immune system does not play a major role in the pathogenesis of DIE lesions. The differences seen in published data on inflammatory processes in endometriosis, and in particular DIE, suggest that further studies might be necessary in order to fully understand the role of these cytokines in this enigmatic disease.

## 4. Materials and Methods

### 4.1. Patients

The plasma and peritoneal fluid (PF) samples were collected as part of the Endometriosis Marker Austria (EMMA) study, a prospective cohort study conducted at the tertiary, certified referral Endometriosis Center. This study was approved by the Ethics Committee of the Medical University of Vienna (EK 545/2010). The premenopausal patients selected for this particular study were between 18 and 50 years of age and all underwent laparoscopic surgery during the period of 2010 to 2015 at the Department of Obstetrics and Gynecology at the Medical University of Vienna, for suspected endometriosis, benign ovarian cysts, uterine fibroids, chronic pelvic pain or infertility. All patients included in the “endometriosis group” all had histologically confirmed endometriotic lesions, whereas patients without endometriosis were defined as the “control group”. During the surgery, the endometriosis cases were scored according the rASRM score as minimal (I), mild (II), moderate (III) or severe (IV). In cases of DIE, the disease was furthermore classified according to the ENZIAN scoring system.

Information on baseline patient characteristics such as age, BMI, gravidity, parity, as well as the intensity of endometriosis-associated symptoms, notably dysmenorrhea and dyspareunia evaluated using the visual analog scale (VAS), was obtained preoperatively. Statistical testing on patient characteristics included Chi-squared tests, parametric t-tests and Mann–Whitney-U test, and was performed using IBM SPSS statistics version 27.

### 4.2. Sample Analysis

Citrate plasma and PF were collected prospectively in accordance with the harmonization guidelines, plasma in a fasting state in the morning on the day of surgery, and peritoneal fluid after a lavage of the peritoneal cavity with 10 mL of sterile 0.9% NaCl. The samples were immediately centrifuged (plasma at 1000 rounds per minute (rpm) and PF at 3000 rpm at 4 °C for ten minutes) and stored in aliquots at −80 °C until analysis.

Ninety-two proteins were measured simultaneously using the “Proseek^®^ Multiplex Inflammation I Panel” (OLINK Proteomics, Uppsala, Sweden). A list of all analyzed proteins is shown in Appendix A.

The measurement was carried out according to the Proseek Multiplex 96 × 96 User Manual. Specifically, the Proseek reagents are based on Proximity Extension Assay technology, in which 96 oligonucleotide-labeled antibody probe pairs bind to their respective protein targets in the sample. A polymerase chain reaction (PCR) reporter is formed by a proximity-dependent DNA polymerization event which is detected and quantified by real-time PCR, generating quantitative values in arbitrary units.

The Fluidigm raw data’s quality was assessed according to OLINK guidelines (Data Preprocessing v1.0). Values in PF were normalized to the total protein amount (measured with the Bradford Assay) in order to compensate for technical differences due to potentially slightly different lavage volumes during PF sample collection.

Biostatistical analysis was performed using BRB Array Tools Version 4.4.1 (developed by the National Cancer Institute, National Institute of Health, Bethesda, United States) and included class comparison (conducted at a significance level of 0.05), whereby different feature selection criteria were applied. The Normalized Protein eXpression (NPX) values were imported into BRB Tools, whereby all normalization methods were disabled, since the data gained in PF were already normalized in the data preprocessing procedure. According to BRB Array Tools software, the log2 transformed data were used for multiplex data analysis, and geometric mean as well as linear fold change between the groups (patients with endometriosis vs. controls and endometriosis with DIE vs. endometriosis patients without DIE) were calculated [58].

## Figures and Tables

**Figure 1 ijms-24-05022-f001:**
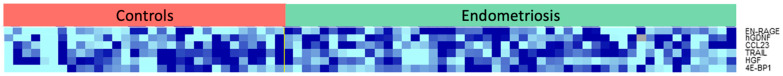
Heat map representing the target proteins that showed a significantly different concentration in plasma of patients with endometriosis (patients with DIE and patients with endometriosis without DIE) compared to controls. The color intensity correlates with the expression level of the protein.

**Figure 2 ijms-24-05022-f002:**
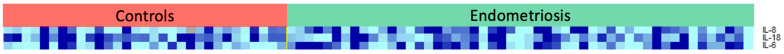
Heat map representing the target proteins that showed a significantly different concentration in peritoneal fluid of patients with endometriosis (patients with DIE and patients with endometriosis without DIE) compared to controls. The color intensity correlates with the expression level of the protein.

**Figure 3 ijms-24-05022-f003:**
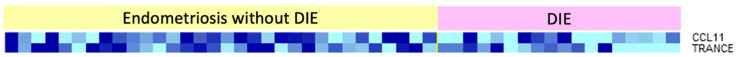
Heat map representing the target proteins that showed a significantly different concentration in plasma of patients with deep-infiltrating endometriosis (DIE) and endometriosis patients without DIE. The color intensity correlates with the expression level of the protein.

**Figure 4 ijms-24-05022-f004:**
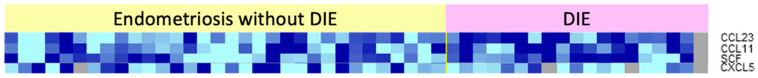
Heat map representing the target proteins that showed a significantly different concentration in PF of patients with deep-infiltrating endometriosis (DIE) and endometriosis patients without DIE. The color intensity correlates with the expression level of the protein.

**Table 1 ijms-24-05022-t001:** Patient characteristics.

Patient Characteristics	Control Group (*n* = 30)	Endometriosis (*n* = 51)	*p*-Value	Endometriosis without DIE (*n* = 33)	Endometriosis with DIE (*n* = 18)	*p*-Value
**General Information**						
Age (years)	34.1 ± 5.97	32.9 ± 6.2	0.434	33.4 ± 5.98	32 ± 6.7	0.536
BMI (kg/m²)	25.2 (21.3–29.2)	21.0 (20.2–24.9)	**0.015**	21.1 (20.2–26.0)	20.9 (19.8–22.9)	0.430
Gravidity	1.5 ± 1.7	0.5 ± 1.1	**<0.001**	0.7 ± 1.3	0.22 ± 0.55	0.171
Parity	0.6 ± 0.7	0.3 ± 0.8	**0.04**	0.45 ± 0.87	0.17 ± 0.51	0.183
**Preoperative pain symptoms**						
Dysmenorrhea *n* (%)	29 (96.7%)	48 (94.1%)	0.609	32 (97%)	16 (88.9%)	0.282
Dysmenorrhea Intensity (VAS)	6.0 (4.0–8.0)	5.0 (5.0–10.0)	**0.024**	8 (5.5–10)	7 (1.75–10.0)	0.562
Dyspareunia *n* (%)	15 (50%)	26 (51%)	0.932	16 (48.5%)	10 (55.6%)	0.629
Dyspareunia Intensity (VAS)	2.0 (2.0–6.0)	0.0 (0.0–6.0)	0.856	0 (0.0–6.0)	3.0 (3.0–5.0)	0.950
Influence of Pain on Sex Life *n* (%)	11 (36.7%)	19 (37.3%)	0.958	11 (33.3%)	8 (44.4%)	0.433
Influence of pain on Sex Life Intensity (VAS)	0.0 (0.0–4.0)	0.0 (0.0–5.0)	0.752	0.0 (0.0–5.0)	0.0 (0.0–4.25)	0.682
**Cycle Phase**			0.620			0.918
Proliferative *n* (%)	13 (43.3%)	25 (49.0%)		16 (48.5%)	9 (50%)	
Secretory *n* (%)	17 (56.7%)	26 (51.0%)		17 (51.5%)	9 (50%)	
**rASRM score** ** *n* ** **(%)**			NA			**0.007**
I	NA	10 (19.6%)		9 (27.3%)	1 (5.6%)	
II	NA	8 (15.7%)		7 (21.2%)	1 (5.6%)	
III	NA	18 (35.3%)		10 (30.3%)	8 (44.4%)	
IV	NA	15 (29.4%)		7 (21.2%)	8 (44.4%)	
Endometrioma *n* (%)	NA	32 (62.7%)	NA	23 (69.7%)	9 (50.0%)	0.164
**ENZIAN score *n*** **(%)**						
A 1–3	NA	13 (25.5%)	NA	NA	13 (72.2%)	NA
B 1–3	NA	17 (33.3%)	NA	NA	17 (94.4%)	NA
C 1–3	NA	6 (11.8%)	NA	NA	6 (33.3%)	NA
FA	NA	0	NA	NA	0	NA
FI	NA	0	NA	NA	0	NA
FO	NA	0	NA	NA	0	NA

Table legend: the statistically significant values are set in bold. DIE: deep-infiltrating endometriosis, BMI: body mass index, VAS: visual analog scale, rASRM: revised American Society of Reproductive Medicine score, NA: not applicable.

**Table 2 ijms-24-05022-t002:** Proteins that were significantly increased or decreased in patients with endometriosis compared to controls.

Target Protein	Geom. Mean of Intensities in		Fold Change	*p*-Value
	Controls (*n* = 30)	Endometriosis (*n* = 51)	in Endometriosis	
PLASMA				
EN-RAGE	10.68	17.44	1.63	0.001
hGDNF	6.20	6.92	1.12	0.05
CCL23	530.17	647.89	1.22	0.03
TRAIL	430.83	385.88	0.90	0.05
HGF	94.14	82.40	0.88	0.03
4E-BP1	50.38	72.20	1.43	0.04
PERITONEAL FLUID				
IL-8	11.34	30.37	2.68	0.04
IL-18	4.36	2.94	0.67	0.01
IL-6	3.68	7.85	2.13	0.03

Table legend: Geom.: geometric, EN-RAGE: Receptor for Advanced Glycation End-products binding protein, CCL23: chemokine ligand 23, HGF: Hepatocyte Growth Factor, 4E-BP1: eukaryotic translation initiation factor 4E-binding protein 1, TRAIL: TNF-Related Apoptosis Inducing Ligand, hGDNF: human Glial-derived Neurotrophic factor, IL-18: Interleukin-18, IL-6: Interleukin-6, IL-8: Interleukin-8.

**Table 3 ijms-24-05022-t003:** Proteins that were significantly increased or decreased in patients with deep-infiltrating endometriosis (DIE) compared to endometriosis patients without DIE.

Target Protein	Geom. Mean of Intensities in		Fold Change	*p*-Value
	Endometriosis without DIE (*n* = 33)	Endometriosis with DIE (*n* = 18)	in DIE	
PLASMA				
CCL11	154.75	130.24	0.84	0.04
TRANCE	24.57	17.11	0.70	0.01
PERITONEAL FLUID				
CCL23	11.18	19.06	1.70	0.004
CCL11	5.52	8.08	1.46	0.03
SCF	7.21	8.85	1.23	0.04
CXCL5	2.57	1.01	2.54	0.046

Table legend. Geom.: geometric, TRANCE: tumor necrosis factor-related activation-induced cytokine, CCL11: chemokine ligand 11, CCL23: chemokine ligand 23, SCF: Stem Cell Factor, CXCL5: C-X-C motif chemokine 5.

**Table 4 ijms-24-05022-t004:** Summary of the characteristics of the target proteins that were significantly increased or decreased in patients with endometriosis compared to controls.

Target	Relation of Factor in Endometriosis Compared to Controls PLASMA	Relation of Factor in Endometriosis Compared to Controls PF	Involved in	Characteristics
Receptor for Advanced Glycation End-products binding protein (EN-RAGE)	↑	NS	Inflammation Oxidative stress	- Associated with multiple inflammatory diseases - Triggers increased production of IL-1ß and TNF-alpha [18] - Secreted by macrophages, lymphocytes and granulocytes [19] - Increases oxidative stress and activation of NF-kB
Chemokine ligand 23 (CCL23)	↑	NS	Chemotaxis Angiogenesis	- Chemoattractant for dendritic cells, resting T-lymphocytes and monocytes - Stimulates cell proliferation and angiogenesis and increases effect of VEGF on vascular endothelial cells [20] - Stimulates production of pro-inflammatory cytokines and adhesion molecules [21]
Hepatocyte Growth Factor (HGF)	↓	NS	Cell proliferation	- Induces cell proliferation, migration and extracellular matrix invasion - Promotes angiogenesis [22] - Said to induce morphogenesis → involved in repair of tissue damage [23]
Eukaryotic translation initiation factor 4E-binding protein 1 (4E-BP1)	↑	NS	Tumor suppression	- Translational repressor - Regulates synthesis of pro-inflammatory cytokines [24] - Inhibits oncogenic factor elF4E (elF4E in turn stimulates cell proliferation) [25]
TNF-Related Apoptosis Inducing Ligand (TRAIL)	↓	NS	Induction of apoptosis	- Ability to bind to 4 different receptors, 2 of which induce apoptosis, 2 are so called “decoy receptors”: compete for TRAIL and neutralize it +/− activate survival pathways via NF-kB, ERP or p38 - Induces apoptosis mostly in “transformed human cells” such as cancer cells - TRAIL expression shown to be downregulated in multiple cancers [26]
human Glial-derived Neurotrophic factor (hGDNF)	↑	NS	Cell growth and survival	- Growth and survival factor - Acts particularly on dopaminergic neurons in substantia nigra and spinal motoneurons - Activity not limited to the nervous system [27]
Interleukin-18 (IL-18)	NS	↓	Inflammation	- Pro-inflammatory cytokine - In the presence of IL-12, enables Interferon-γ production in, amongst others, T-helper-1 cells, NK cells and macrophages - In the presence of IL-2, enables cytokine production in CD4+ NKT cells, NK cells and T-helper-1 cells [28]
Interleukin-6 IL-6	NS	↑	Immune response	- Effects on immune response and hematopoiesis - Induces production of acute-phase proteins → increased production of T-helper cells and cytotoxic T cells [29,30] - In cancer cells: described as regulating anti-apoptosis, cell survival, proliferation and invasion [30]
Interleukin-8 (IL-8)	NS	↑	Immune response Angiogenesis	- Chemokine → secreted by macrophages and monocytes - Chemotactic agent for neutrophils, as well as for a subset of T-lymphocytes - Stimulates angiogenesis [31]

Table legend—↑: increased, ↓: decreased, NS: not significant, PF: peritoneal fluid, IL-1ß: Interleukin-1ß, TNF-alpha: Tumor Necrosis Factor-alpha, NF-kB: Nuclear Factor kappa B, NKT: natural killer T-cells, VEGF: Vascular Endothelial Growth Factor, ERP: Exported Repeated Protein, IL-2: Interleukin-2.

**Table 5 ijms-24-05022-t005:** Summary of the characteristics of the target proteins that were significantly increased or decreased in patients with deep-infiltrating endometriosis (DIE) compared to endometriosis patients without DIE.

Target	Relation of Factor in Patients with DIE Compared to Endometriosis Patients without DIE PLASMA	Relation of Factor in Patients with DIE Compared to Endometriosis Patients without DIE PF	Involved in	Characteristics
Tumor necrosis factor-related activation-induced cytokine (TRANCE)	↓	NS	Adaptive immunity	- Regulates dendritic- and T-cell interaction - Supports dendritic-cell survival [45] - Involved in bone remodeling [46]
Chemokine (C-C Motif) ligand 11 (CCL 11)	↓	↑	Allergic reaction	- Produced by wide range of cells such as macrophages, T and B cells, epithelial cells, etc. - Recruits eosinophiles to inflammatory cites during allergic reaction - Increased levels described in multiple neuro-inflammatory and neurodegenerative disorders [47]
Chemokine ligand 23 (CCL23)	NS	↑	Inflammation	- Chemoattractant for dendritic cells, resting T-lymphocytes and monocytes - Stimulates cell proliferation and angiogenesis ad increases effect of VEGF on vascular endothelial cells [20] - Stimulates production of pro-inflammatory cytokines and adhesion molecules [21]
Stem Cell Factor (SCF)	NS	↑	Innate immunity	- Involved in regulation of proliferation, differentiation, migration and apoptosis, notably in mast cells [48] - May affect morphological changes and adhesions of mast cells → involved in innate immunity [49]
C-X-C motif chemokine 5 (CXCL5)	NS	↑	Adaptive immunity Angiogenesis	- Plays a role in angiogenesis and local recruitment of B and T cells and eosinophils - Overexpressed in multiple malignant tumors such as endometrial cancer and breast cancer - In hepatocellular carcinoma, expression correlated with degree of metastatic potential, inflammatory infiltration and malignancy [50]

Table legend—↑: increased, ↓: decreased, NS: not significant, PF: peritoneal fluid, VEGF: Vascular Endothelial Growth Factor.

## Data Availability

The authors confirm that the data supporting the findings of this study are available within the article.

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
