# Peer review of "Does the Use of the “Proseek^®^ Multiplex Inflammation I Panel” Demonstrate a Difference in Local and Systemic Immune Responses in Endometriosis Patients with or without Deep-Infiltrating Lesions?"

_ijms, 2023, doi:10.3390/ijms24055022_

Round 1

Reviewer 1 Report

Dear authors,

I find the article well written. The study is interesting and I find that it enriches the literature of endometriosis, which unfortunately still has many gaps, especially about the pathophysiology of the disease.

Reviewer 2 Report

I read with great interest the manuscript, which falls within the aim
of this Journal. In my honest opinion, the topic is interesting enough
to attract the readers’ attention. Nevertheless, authors should clarify
some points and improve the introduction. Authors should consider the
following recommendations:

- Manuscript should be further revised in order to correct some typos
and improve style.

- Introduction part. It would be appreciable a small topic on management
of deep endometriosis (referring to PMID 33687160) and its correlations
with infertility and pregnancy outcomes. Moreover an insight into how this pathology can affect
the quality of life of patients (referring to doi:
10.1186/s10397-021-01096-5)

- Table 1. Explain better what you mean for "influence of pain in sex
life" and why this should be important to your study.

Reviewer 3 Report

Perricos et al. presented some interesting data to changes in the immune system in endometriosis. In summary, they concluded that DIE is a stable disease. However, I have some concerns about this far-reaching conclusion which are detailed below.

Major Points

Gordts et al. in Fertil Steril 2017;108:872 presented several interesting aspects to the pathogenesis of endometriosis (especially of DIE) and asked whether endometriosis is really a progressive disease and thus should be discussed in the current study more in depth. Especially with respect to the main conclusion of the authors that DIE is a stable disease, I really wonder which data presented the authors to substantiate their conclusion. In my opinion progression and/or stability of a disease can only be analyzed by repeated histological monitoring of the lesions at several different time points or sometimes with MRI and/or ultrasound. In contrast, the authors only presented data showing that most of the cytokines studied in cases of DIE did not differ significantly from healthy cases. Maybe this might be due to other reasons, e.g. better blockade of the immune system by immune-suppressive cytokines. Possibly DIE lesions block the immune system via TGF-beta1 or something else. Furthermore, if we look at the conclusion of the authors presented in J Clin Med 2020;9:1 [ref. 58] that DIE is a unique entity with enhanced angiogenetic and pro-proliferative features, the authors should have discussed both conclusions more exhaustive, e.g. is a disease with increased angiogenetic and pro-proliferative properties really stable?        

I would also have liked the involvement of the immune system in the pathogenesis of endometriosis to be discussed in more detail. Although we still do not know exactly whether changes in the immune system are a cause or a consequence of the disease, there is evidence from animal experiments for a causal effect (e.g. Burns et al., 2018, Endocrinology 159:103).

Table 3 – CCL11 and TRANCE are decreased in the plasma of DIE patients. However, why did the authors not calculate the PPV, NPV, sensitivity, specificity etc. in order to evaluate whether both proteins might by suitable for a non-invasive diagnostic test?

In my opinion the title is not describing the main finding of the study but instead is focused on the suitability of the use of the Proseek Multiplex Inflammation I Panel to study immune responses in endometriosis. Nowhere the results obtained have been validated by e.g. ELISAs with plasma samples from patients which could and should have been done.

Minor Points

1.       Tables 2 and 3 – please explain the meaning of Geom

2.       Table 2 – please correct the typo 112 to 1.12 and please give the exact p-value for EN-RAGE, 0.00 is not enough.

3.       Lines 123-130 and 155-158 – please replace all commas in –fold by dots.

4.       Table 3 legend – why is CXCL5 given in italics?

5.       Discussion, line 164 – Why do you mention the exact pathogenesis, the word exact is not necessary

6.       Table 4 and 5, right panel – sometimes small letters and sometimes upper case letters are used.

7. Table 5 – please replace pateints by patients

Round 2

Reviewer 3 Report

The authors have addressed all my concerns and have made the appropriate changes.